# Extensive public health initiatives drive the elimination of *Aedes aegypti* (Diptera, Culicidae) from a town in regional Queensland: A case study from Gin Gin, Australia

**Brendan J. Trewin**[1]*, **Brian L. Montgomery**[2], **Tim P. Hurst**[2], **Jason S. Gilmore**[2], **Nancy M. Endersby-Harshman**[3], **Greg J. Crisp**[2]

**1** CSIRO, Health & Biosecurity, Brisbane, Queensland, Australia, **2** Queensland Health, Brisbane, Queensland, Australia, **3** University of Melbourne, Bio21 Institute, Parkville, Victoria, Australia

* Brendan.trewin@csiro.au

## Abstract

*Aedes aegypti* is the primary vector of exotic arboviruses (dengue, chikungunya and Zika) in Australia. Once established across much of Australia, this mosquito species remains prevalent in central and northern Queensland. In 2011, *Ae. aegypti* was re-discovered in the town of Gin Gin, Queensland, by health authorities during routine larval surveillance. This town is situated on a major highway that provides a distribution pathway into the highly vulnerable and populous region of the state where the species was once common. Following the detection, larval habitat and adult control activities were conducted as a public health intervention to eliminate the *Ae. aegypti* population and reduce the risk of exotic disease transmission. Importantly, genetic analysis revealed a homogenous cluster and small effective population vulnerable to an elimination strategy. By 2015, adult surveillance revealed the population had expanded throughout the centre of the town. In response, a collaboration between research agencies and local stakeholders activated a second control program in 2016 that included extensive community engagement, enhanced entomologic surveillance and vector control activities including the targeting of key containers, such as unsealed rainwater tanks. Here we describe a model of the public health intervention which successfully reduced the *Ae. aegypti* population below detection thresholds, using source reduction, insecticides and novel, intensive genetic surveillance methods. This outcome has important implications for future elimination work in small towns in regions sub-optimal for *Ae. aegypti* presence and reinforces the longstanding benefits of a partnership model for public health-based interventions for invasive urban mosquito species.

## Author summary

The yellow fever mosquito, *Aedes aegypti*, has rapidly emerged as the primary cause of considerable morbidity and economic loss across the tropics, due to its high competency

**Data Availability Statement:** All relevant data are within the manuscript and its Supporting Information files.

**Funding:** NMEH was supported by National Health and Medical Research Council program grants 1037003 & 1132412. https://www.nhmrc.gov.au/ The funders had no role in study design, data collection and analysis, decision to publish, or preparation of the manuscript.

**Competing interests:** The authors have declared that no competing interests exist.

for transmitting arboviral diseases to human populations. Australian heath authorities have a long history of successfully eliminating populations of this species with traditional forms of mosquito control in areas sub-optimal for presence. Here we document the elimination of a population of *Ae. aegypti* from Gin Gin, Australia, through an extensive and locally resourced public health initiative which included the use of community engagement, traditional mosquito control, population genetics and novel surveillance tools. The documentation and outcomes presented here provide insight into modern suppression campaigns and may inform health authorities on strategies for future invasive mosquito incursions and elimination of isolated *Ae. aegypti* populations.

## Introduction

*Aedes aegypti* (Linnaeus, 1762) (Diptera: Culicidae) is the primary vector of dengue fever, chikungunya and Zika on mainland Australia. Globally, *Ae. aegypti* is responsible for the rapid re-emergence and spread of viral diseases over the past 40 years [1]. Currently, the number of annual global dengue cases is estimated to be 390 million [2], while the Zika epidemic in South America during 2015 was largely attributed to this species [3]. *Aedes aegypti* exhibits several characteristics which make it one of the most invasive of all mosquito species and a competent vector of dengue. These include a lifecycle highly adapted to urban environments including; a penchant for human blood with multiple feedings per gonotrophic cycle, desiccation resistant eggs that can survive dry and winter conditions, and the utilization of domestic containers as larval habitat [4]. Historically, these traits have enabled the species to spread widely within Australia, causing epidemics of dengue fever which have shaped public health policy [5].

*Aedes aegypti* is postulated to have been introduced into Australia around the time of British colonization, primarily through water storage on large sailing ships [6]. By the early 1900s it had spread throughout the eastern seaboard as far south as the Victorian border [5,7]. During the early part of the 20th century, *Ae. aegypti* was responsible for large epidemics of dengue fever in Queensland, with some affecting up to 90% of populations in major urban centres such as Brisbane [7]. During the mid-1900s, the *Ae. aegypti* distribution subsequently declined into central and northern Queensland, where temperature and rainfall conditions are favourable for the species [5,8]. With a rapid increase in dengue prevalence worldwide since the 1980s, Queensland observed a resurgence of the disease in northern regions of the state [9]. The artificial introduction and replacement of *Ae. aegypti* populations with a *Wolbachia* symbiont in 2010 [10] has subsequently reduced the threat of autochthonous dengue outbreaks in northern Queensland by blocking viral transmission within adult mosquitoes [11].

Traditional *Ae. aegypti* management focuses on the removal of larval habitat and the application of insecticides to either control adults or larvae [12]. In Australia, the distribution of *Ae. aegypti* has retracted into central Queensland following the adoption of reticulated water and intensive post-World War 2 public health interventions that removed rainwater tanks (a permanent source of larval habitat) [5,8]. Health authorities targeting larval habitat in the state's capital, Brisbane, led to the elimination of *Ae. aegypti* around 1957 and cessation of local dengue transmission since 1947 [5]. More recently, the elimination of incursions of *Ae. aegypti* populations from Queensland into the Northern Territory (Tennant Creek and Groote Eylandt) were undertaken through effective community education, larval source reduction and targeted insecticide treatments and monitoring programs [13,14]. These same processes form the core strategy developed by the Queensland government to manage dengue transmission [12].

Central and southern Queensland regions remain vulnerable to exotic disease outbreaks where the wild-type *Ae. aegypti* (no *Wolbachia* infection) vector is abundant and interacts with infective travellers. During mid-2019, Rockhampton recorded the first dengue outbreak in over 60 years [15,16]. It is unknown whether areas outside north Queensland are suitable for *Wolbachia* introduction [17,18]. Thus, traditional methods of mosquito control remain the strategy to reduce the risk of disease transmission by health authorities in southern Queensland regions [12]. South East Queensland (SEQ) is highly vulnerable to the establishment of *Ae. aegypti* [19] and receives a high proportion of the annual numbers of viraemic travellers to Queensland [20]. Developing regional capacity and capability alongside innovative strategies to eliminate invasive mosquito populations is particularly important when minimizing the risk of exotic disease transmission in large urban centres.

In 2011 *Ae. aegypti* was re-discovered in the small town of Gin Gin after 25 years through house-to-house surveys by state health authorities. The species was not identified in small surveys in 1996 and 2006 (B. Montgomery, Queensland Health, pers. comm.). Here we provide a public health perspective and document the extensive and enhanced entomological surveillance and control activities required to eliminate *Ae. aegypti* from Gin Gin that serve as a calibration exercise for similar or larger-scale interventions.

## Methods

### Ethics statement

Human ethics approval for the 2015–2016 community engagement, mosquito suppression and research was provided by the CSIRO Health and Medical Research Human Research Ethics Committee (Proposal #12/2015) and QIMR Berghofer Human Ethics Committee (#P2054).

Chronology of intervention and the various annual activities are described below.

### Summer 2011–2012. Initial detection, Surveillance, Control and community engagement

Gin Gin (24.9908˚ S, 151.9500˚ E) is a small town (1,053 population) [21] in the Wide Bay Burnett region of Queensland, Australia, located on a major highway into the state capital of Brisbane (Fig 1). In Queensland, local governments are responsible for monitoring and enforcing the *Public Health Act 2005* [22] and *Public Health Regulation 2018* [23]. These acts give local government the power to enter premises without consent for both mosquito surveillance and control, dependent upon the threat to the community (mosquito surveillance or prevention of disease transmission). Following the detection of *Ae. aegypti* in Gin Gin during 2011, a report was provided to the Bundaberg Regional Council (BRC) by Queensland Health. In order to prevent the regional spread of the species, this report recommend:

1. Elimination of the Gin Gin *Ae. aegypti* population to reduce the risk of dengue transmission (from viraemic travellers).

2. Surveying premises in which *Ae. aegypti* were detected (at least once a month for five months during summer), to determine whether *Ae. aegypti* was still present.

3. Surveying premises adjacent to those where *Ae. aegypti* were present to ensure these were also free of the species.

Bundaberg Regional Council conducted routine adult surveillance at and around the premises where *Ae. aegypti* was first collected (Fig 2A). One oviposition trap (ovitrap) containing aged rainwater, a lucerne pellet and an oviposition substrate (typically a red material or water-

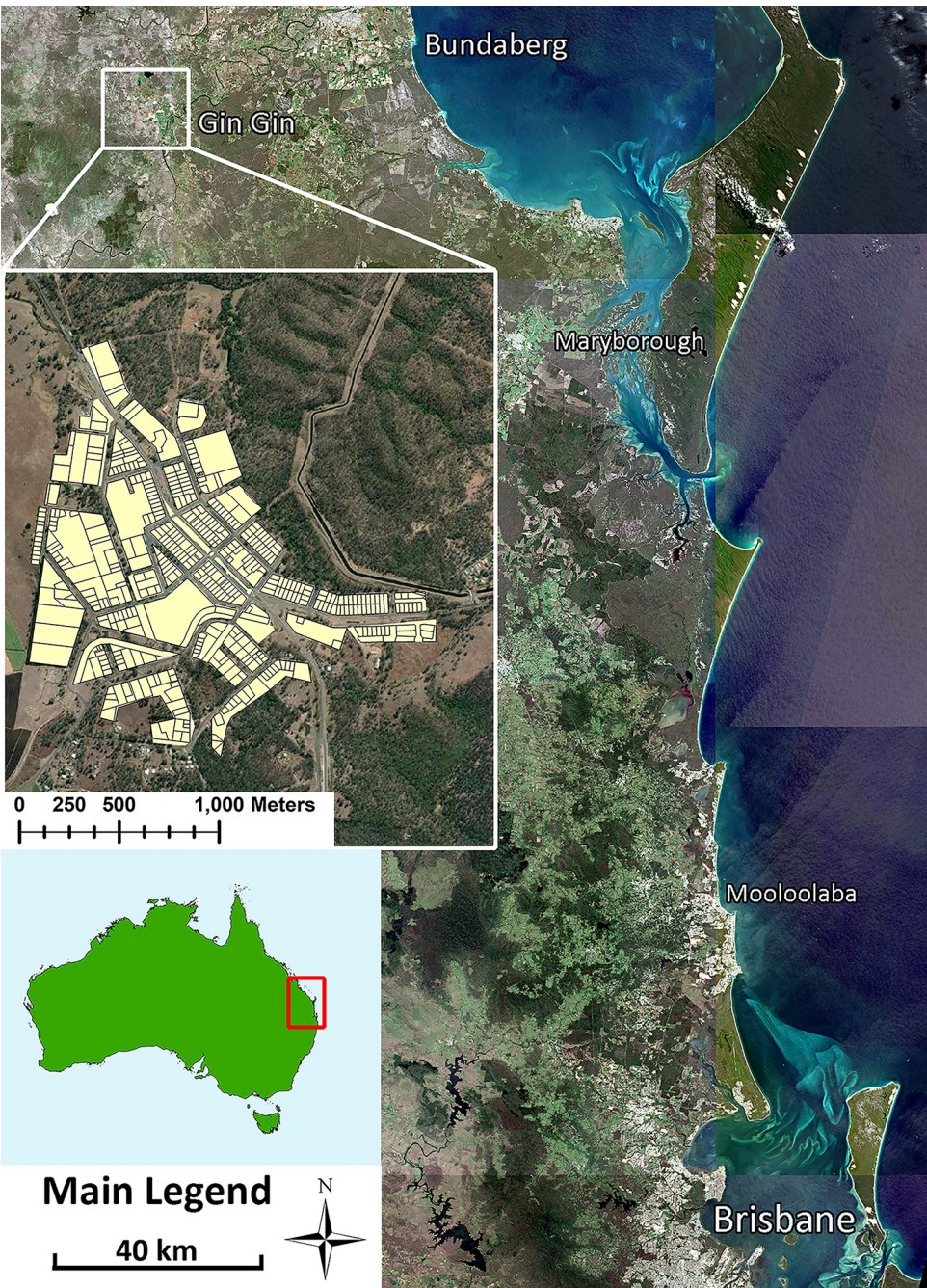

**Fig 1. Location of the town of Gin Gin relative to Brisbane and South East Queensland.** Location within Australia (bottom left) and Gin Gin town layout including scale (middle left). Map Source: Base Layer assembled from the Open Access Copernicus Australasia Regional Data Hub [27,28] and Australian map and residential features digitized from public domain cadastre data [29,30] in ArcGIS.

proof sandpaper (80 grit) (S1 Text) [24] and BG Sentinel trap (BGS; Biogents GmbH, Regensburg, Germany) [25] were set at the original positive premise, the surrounding properties and other high-risk locations (caravan park, ambulance station, and council depot). Ovitraps were collected weekly between the 4th and 11th of May and BGS traps between the 4th of May and 25th of October 2011. Eggs from ovitraps reared to fourth instar larvae for identification to

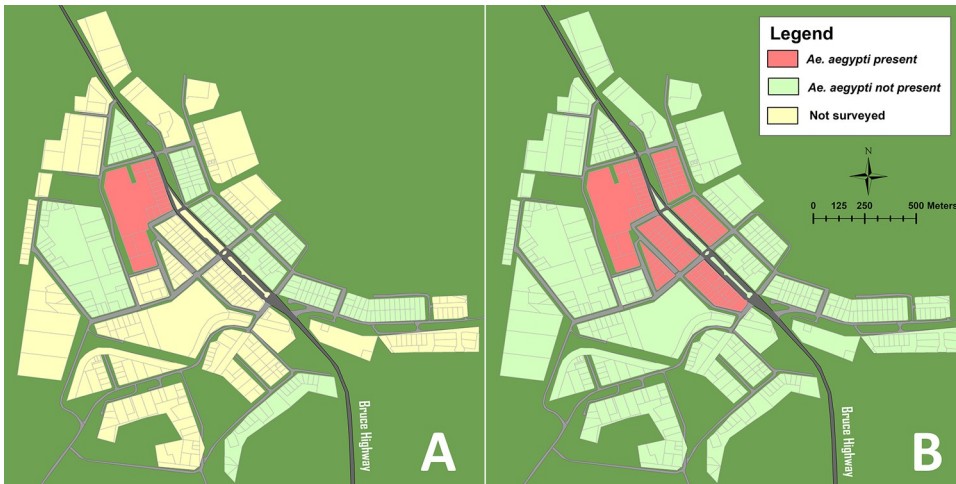

**Fig 2. Detection block 2011 (A) and town-wide larval surveys 2012 (B) of *Aedes aegypti* in Gin Gin.** Red indicates blocks where *Aedes aegypti* were detected, green where no *Ae. aegypti* were found to be present, and yellow blocks were not surveyed. Base layer imagery digitized manually from public domain cadastre data [29,30].

species via microscopy and key [26]. A population suppression program was implemented with BRC the lead agency and technical support provided by Queensland Health. Council staff were trained in house-to-house inspection methodologies and larval surveillance techniques prior to an extensive larval survey (473 premises). Larval surveillance and control was undertaken over a two week period (30th January to 17th February 2012) with methods documented in S1 Text.

Briefly these methods were; verbal consent was obtained to conduct an inspection at each premises and if nobody was present the property was re-visited. Premises were scored by the Premise Condition Index (PCI) [31] and all natural and artificial containers holding water were categorised into groups of breeding sites via the Barker-Hudson method [32] and checked for the presence of mosquito larvae. A sub-sample (6 to 12 larvae) from each positive larval habitat was placed into 80% ethanol for species identification by microscopy. A second round of adult surveillance was undertaken by BGS traps set fortnightly (10 premises, 22nd March– 14th May 2012) and ovitraps set weekly (14 premises; 21st March– 22nd May 2012).

Legal mechanisms in Queensland were activated to enable this intervention. Firstly, an 'Authorised Inspection Program' was implemented under the *Local Government Act 2009* [33] to grant powers of entry to yards (not inside houses) by authorised officers without a resident's consent. Secondly, all chemical treatment was consistent with label recommendations, conducted by or supervised by a licensed Pest Management Technician (PMT) and a *Pest Control Advice* (PCA) provided for each premises when treatment occurred. Larval control activities consisted of the removal and/or insecticide treatment of containers that contain or have the potential to contain *Ae. aegypti*. Prolink Pellets containing the insect growth regulator (*S*)-methoprene were applied to containers that were either; hard-to-inspect, or were large or could not be emptied or removed (e.g. drain sumps, drums, tyres, tree holes). Prolink (*S*)-methoprene ProSand was applied to leaf axils of bromeliads that retain water. To prevent the emergence of adult mosquitoes from large permanent water sources, Prolink (*S*)-methoprene XR Briquets were used in damaged rainwater tanks or if their screens had been removed. Adulticides are not used in the region for mosquito control, however, to target adult *Ae. aegypti* two lethal ovitraps [34] treated with Bifenthrin were set at each of the positive premises (S1 Text).

As part of the 2012 elimination strategy, a community engagement plan was established through targeted awareness campaigns and community engagement strategies including:

- a 'Survey to eliminate *Aedes aegypti* to reduce a public health risk' fact sheet,

- a media release on the 'Gin Gin *Ae. aegypti* elimination program',

- letter for premises that were positive for *Ae. aegypti*,

- letter for premises within a 100 m radius of premises that were positive for *Ae. aegypti* and,

- an information sheet regarding prevention and control of mosquito breeding in the yard.

A full description of 2012 surveillance, control and community engagement methods during the first suppression program is documented in S1 Text.

## 2012–2013. Genetic assessment

A population genetics study was undertaken by BRC to understand the likelihood of *Ae. aegypti* population elimination following removal of the extant population. Samples of larvae (n = 39) collected from containers during house-to-house surveys (April 2013) were sent to University of Melbourne (Pest and Environmental Adaptation Research Group) and assessed at the genetic level for population structure using neutral microsatellite markers and compared with other published data on samples from central Queensland [35]. Allelic richness calculated from 15 individuals to match the size of the smallest population in the central Queensland study (A) [35], gene diversity (He), pairwise $F_{ST}$ and inbreeding coefficient ($F_{IS}$) were estimated using FSTAT 2.9.3.2. Mean effective population size (Ne) was estimated using ONe-Samp 1.2. A Bayesian analysis to estimate the number of populations within the sample data was made using STRUCTURE (Version 2) [36]. A burn-in length of 100, 000 was chosen followed by 250, 000 iterations and the simulation was run using the admixture model with allele frequencies uncorrelated among populations. The number of populations within the data ($K$) is estimated by checking the fit of the model for a range of $K$ values. $K$ values of 1 to 8 were tested with five runs for each value of $K$. We used the method of Evanno *et al.* [37] to estimate the true $K$ as applied in STRUCTURE Harvester. To identify first generation immigrants, we used GeneClass2 [38] with a Bayesian method [39] for detecting first-generation immigrants with the resampling algorithm of Paetkau et al. [40] with 10000 simulated individuals, α = 0.01 and the L_home likelihood computation.

## 2013–2014. Surveillance activities

A larval and adult trapping survey was undertaken by Queensland Health (February to April 2014). House-to-house surveys were undertaken at 73 premises, and Gravid *Aedes* Traps (GAT; Biogents GmbH, Regensburg, Germany) [41] were placed within six of these for 13 days and serviced after seven days.

## 2014–2015. Surveillance activities

A rainwater tank survey on eight rainwater tanks in central Gin Gin premises was performed in December 2014. Surveys include tank compliance with state health regulations [23] and a larval sample was taken from each rainwater tank via the methods of Knox *et al.* [42] where a large sweep net is used to exhaustively sample a large volume of water. Three ovitraps were set around unsealed rainwater tanks for four weeks.

An oviposition study (February—mid-April 2015) was undertaken at nine premises for ten weeks. A single GAT and four ovitraps were placed within the yard of each premises. Traps were serviced fortnightly, eggs counted, and larvae reared to fourth instar for identification to species via microscopy and key [26].

## 2015–2016. Surveillance, Control and community engagement

A second *Ae. aegypti* suppression program was undertaken in conjunction with a mark-release-recapture (MRR) study in Gin Gin during 2016 and reduce the probability that released *Ae. aegypti* females would increase the risk of disease transmission [43]. Forty premises and 58 rainwater tanks were surveyed with the adult trapping methodology and rainwater tank non-compliance survey documented in Trewin *et al*. [43]. In this survey, rainwater tanks were examined for their compliance with state regulations (sealed, with mesh on both exit and entry) [23]. During the MRR, locally caught adult *Ae. aegypti* and *Ae. notoscriptus* (Skuse) were released to study the movement of vectors between urban landscape features such as rainwater tanks, and recaptured in a network of BGS and GAT traps [43]. As part of this study, extensive efforts were made to engage and educate the local community on the risk of *Ae. aegypti* presence on their premises (S2 Text). Community engagement included the formation of a community reference group, town hall meeting, educational flyers and media activities. Homeowners were encouraged to clean-up surplus containers from yards and seal rainwater tanks. During this second *Ae. aegypti* suppression program, all non-compliant rainwater tanks as defined by regulatory standards (e.g. mesh size apertures no greater than 1 mm to prevent entry or egress of mosquitoes) [23] were sealed with new mesh screens or silicone in rust-related holes. Tanks unable to be sealed were treated with residual insecticides (Prolink XR Briquets) and residents were encouraged to decommission high risk tanks. Traditional mosquito control was undertaken as part of the risk mitigation strategy including source reduction, residual insecticide treatments of Prolink ProSand in bromeliads, and Prolink XR Briquets in larger containers. Indoor residual spraying (IRS) [44] with Bifenthrin was offered to residents once the experiment was complete (with two residents and one business opting to have their premises treated). Residents were compensated for their participation through an inexpensive voucher system for redeeming local produce. For the 2015–16 suppression program community engagement, risk management and mosquito population suppression plans see S2 Text, S3 Text and S4 Text, respectively.

## 2017–2018. Surveillance activities

Records for surveillance by BRC during 2016–2017 were unavailable for analyses. During summer 2017–2018 adult surveys using BGS traps for five weeks across five premises. Adults were classified adults to species by microscope and key [26].

## 2018–2019. Surveillance activities

As part of a larger regional population genomics survey [45], BGS and ovitrapping was undertaken in Gin Gin for ten weeks (February until May 2018), in six houses previously positive to *Ae. aegypti*. A single BGS trap and four ovitraps were placed in the yard of each house and serviced weekly. Concurrent BGS surveillance was undertaken by BRC (five weeks across five premises) in late summer. Larvae (reared to 4th instar) and adults were identified to species via microscope and key [26].

## 2019–2020. Presence-absence surveillance: Rapid surveillance for vector presence survey

To interrogate the detection threshold indicated by negative records from the previous three summers, a highly sensitive *Ae. aegypti* survey was conducted using an innovative method that links ovitrap samples to molecular diagnostics. Rapid Surveillance for Vector Presence (RSVP) [46] can rapidly detect *Ae. aegypti* nucleic acids by using real-time reverse transcription

polymerase chain reaction (RT-PCR) to screen large amounts of genetic material. Large volumes of endemic species can be processed by aggregating egg samples in cohorts (<5000 eggs) that typically are sourced from multiple ovitraps. The sensitivity of RSVP facilitates the efficiency of a regional presence-absence survey of target invasive species over large spatial and temporal scales, particularly when they are expected to be absent or in very low numbers. Since 2017, RSVP has been offered to regional councils in and near SEQ by Queensland Health on a seasonal basis. Premises across the town were selected to ensure all high-risk residential blocks within Gin Gin were sampled for the presence of *Ae. aegypti*. Twenty-one premises were surveyed with a single ovitrap placed within the yard and eggs collected fortnightly for two periods of four weeks (total 8 weeks from February until March 2020).

## Mapping

Trapping and larval surveillance results are mapped at the block scale (Gin Gin residential blocks range from 0.39 to 15.20 ha), the spatial unit under which *Ae. aegypti* is optimally targeted due to limited dispersal abilities [47], while also preserving the privacy of individual premises where surveillance was undertaken. All shapefile maps were digitized in ArcGIS by outlining residential features (blocks, roads, highways) and then overlaying public domain cadastre data [29,30]. Base layer imagery of South East Queensland region sourced from the open access Copernicus Australasian Regional Data Hub [27,28]. Trap days is a quantitative measure of the number of traps placed in the environment multiplied by the number of days present when the population was surveyed over a summer season (November until May) [48].

## Results

### 2011–2012. Initial detection and first suppression program

*Aedes aegypti* was first detected in a single property in central Gin Gin by a routine Queensland Health house-to-house survey in 2011 (Fig 2A). Mosquito surveillance and suppression activities subsequent to detection revealed *Ae. aegypti* in 2.3% (11/473) of premises throughout central Gin Gin (Fig 2B), representing a modest increase from the 1986 detections (3 premises). During larval surveys, a total of 5,035 wet and 724 dry containers were detected, an average of 12 larval habitat sites per premises (Table 1). Mosquito larvae (Table D in S1 Text) were present in approximately 40% and 11% of all premises and wet containers, respectively (Table 1). The most prevalent container category positive for *Ae. aegypti* (35%) were garden accoutrements such as plant pots and saucers, birdbaths, buckets and striking pots (Table E in

**Table 1. Prevalence of water bearing containers and number with *Aedes aegypti* juvenile stages in Gin Gin, Australia.** Container data collected during town-wide surveillance activities summer 2011–12 and classified by the Barker-Hudson method [32].

| Category | Total (%) | Wet (%) | Wet containers with *Ae. aegypti* present (%) |
|---|---|---|---|
| Garden Accoutrement | 1,796 (31.2) | 1,408 (28.0) | 6 (35.3) |
| Discarded Household Item | 460 (8.0) | 409 (8.1) | 4 (23.5) |
| Domestic Use Container | 259 (4.5) | 246 (4.9) | 4 (23.5) |
| Recreational Item | 63 (1.1) | 62 (1.2) | 1 (5.9) |
| Water Storage | 162 (2.8) | 157 (3.1) | 1 (5.9) |
| Rubbish | 200 (3.5) | 159 (3.2) | 0 |
| Building Fixture | 224 (3.9) | 205 (4.1) | 0 |
| Natural Habitat | 2,248 (39.0 | 2,044 (40.6) | 0 |
| **Total** | **5,759** | **5,035** | **17** |

**Table 2. Summary of control activities during the first (2011–2012) and second (2015–2016) suppression programs.** Includes larval habitat treatment with juvenile hormones (S-methoprene), source reduction (sealing, removal) and adult control (Bifenthrin).

| | | Larval Habitat (Containers) | | Larval Habitat (Containers) | | Adult Control | |
|---|---|---|---|---|---|---|---|
| | | S-methoprene Treatment | | Reduction | | Insecticide Treatment | |
| Suppression | Premises | Pellets | Briquette | Rainwater | Containers | Bifenthrin | Number of Lethal Ovitraps |
| Program | Treated | (bromeliads) | Treatment | Tanks Sealed | Removed | IRS | Ovitraps Deployed (weeks) |
| 2011–2012 | 151 | 1033 (502) | 24 | 8 | 17 | 0 | 22 (8) |
| 2015–2016 | 13 | 40 (21) | 1 | 24 | 44 | 4 | 0 |

S1 Text). Three hundred and forty-seven rainwater tanks (73.4% of premises inspected) were recorded during the survey, with most tanks containing water. Due to difficulty of access, not all tanks were inspected or sampled. Four tanks were positive for mosquito larvae and one was positive for *Ae. aegypti* larvae. Premises from the initial 2011 larval surveys for *Ae. aegypti* were positive in one of the 14 ovitrap locations and one of three BGS trap locations over the same ten-week period in 2012 (Fig 2).

Control activities included treating wet larval habitat sources such as bromeliad axils (503 total) and rainwater tanks (9 total) with S-methoprene pellets and briquettes, respectively (Table 2). Eight rainwater tanks were sealed, and 17 rubbish-containers removed (Table 2).

## 2012–2013. Surveys and genetic assessment

*Aedes aegypti* was present across six blocks in central Gin Gin, collected from yard containers and one rainwater tank (Fig 3 and Table A in S1 Table). The thirty-nine samples collected showed the lowest degree of allelic richness (i.e. low number of microsatellite alleles adjusted for sample size) and lowest gene diversity of the samples tested, which included a range of locations in central and northern Queensland (Table 3). The inbreeding coefficient for the Gin Gin sample was moderate, but significant (Table 3) and no first-generation immigrants were detected. Pairwise FST estimates between sample localities revealed significant population differentiation between the samples of *Ae. aegypti* from Gin Gin and all other localities (Table 4). The only samples not significantly differentiated from each other were from Gordonvale and Yorkeys Knob.

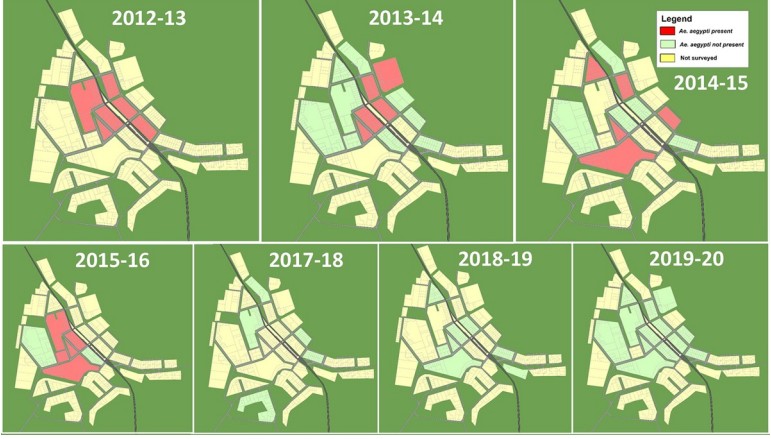

**Fig 3. Annual summer surveys of residential blocks surveyed for *Aedes aegypti* in Gin Gin, Australia between 2012-and 2020.** Red colouring indicates a block where *Ae. aegypti* was surveyed as present, green where surveillance occurred but did not detect the species and yellow where surveillance was not undertaken. Base layer imagery digitized manually from public domain cadastre data [29,30].

**Table 3. Genetic diversity over seven microsatellite loci for *Aedes aegypti* from eleven locations in Queensland, Australia.** Results grouped in regions (north Queensland—*NQL*, and central Queensland—*CQL*) with: sample size (N), allelic richness calculated from 15 individuals (A), gene diversity (He), inbreeding coefficient ($F_{IS}$), mean effective population size (Ne) and 95% confidence limits (in parentheses) (Table modified from Rašić *et al.* [35] to include data from Gin Gin).

| Name | Code | Latitude | Longitude | N | A | He | $F_{IS}$ | Ne |
|------|------|----------|-----------|---|---|-----|---------|-----|
| *North QLD* | | | | | | | | |
| Yorkeys Knob | 1 | -16.8094 | 145.7226 | 30 | 3.38 | 0.544 | 0.125 | 43 (22–146) |
| Gordonvale | 2 | -17.0966 | 145.7787 | 28 | 3.15 | 0.494 | 0.164 | |
| Ingham | 3 | -18.6533 | 146.1604 | 15 | 3.00 | 0.519 | 0.083 | 15 (10–37) |
| *Central QLD* | | | | | | | | |
| Rockhampton | 4 | -23.3795 | 150.4995 | 30 | 3.13 | 0.479 | 0.134 | 33 (19–101) |
| Mt Morgan | 5 | -23.6449 | 150.3889 | 34 | 2.75 | 0.442 | 0.108 | 19 (10–57) |
| Duaringa | 6 | -23.7110 | 149.6710 | 29 | 3.19 | 0.457 | 0.079 | 24 (14–72) |
| Bluff | 7 | -23.5786 | 149.0703 | 32 | 2.78 | 0.477 | 0.054 | 25 (15–90) |
| Emerald | 8 | -23.5162 | 148.1610 | 28 | 3.23 | 0.426 | 0.054 | 65 (33–282) |
| Capella | 9 | -23.0837 | 148.0245 | 21 | 2.98 | 0.461 | -0.078 | 22 (14–63) |
| Longreach | 10 | -23.4433 | 144.2509 | 28 | 3.07 | 0.443 | -0.063 | 27 (17–105) |
| *Gin Gin* | 11 | -24.98946 | 151.9500 | 39 | 2.51 | 0.329 | 0.093 | 19 (14–28) |

Effective population size in Gin Gin, estimated by ONeSamp, was small and similar to most samples from central Queensland (Table 2; mean = 19.21, median = 19.15, lower 95% CL = 13.58, upper 95% CL = 28.04). STRUCTURE analysis gave an estimate of six genetic clusters (K) within the complete dataset from Queensland, using both the highest log probability of the data and the ΔK method [34] (Table 2 and Table B in S1 Table). For K = 6, all locations showed some degree of admixture, but for Gin Gin, admixture was minimal (Figure A in S1 Table). Yorkeys Knob and Gordonvale from north Queensland were grouped together; Longreach, Bluff, Duaringa and Emerald were separate clusters, while other locations showed a high degree of admixture (Figure A in S1 Table).

## Summary of surveillance and control activities 2013–2020

Six positive premises were detected in 2013–2014 (five from larval surveys and from one GAT at a separate premises, across five residential blocks in north-eastern Gin Gin (Fig 3). Six of eight rainwater tanks inspected were non-compliant with regulations, but not sampled for *Ae. aegypti*. Larval surveys (2014–2015) suggest *Ae. aegypti* distribution had expanded southward in residential blocks, with five of 10 blocks positive (Fig 3). The most extensive trapping effort (applying GAT and ovitraps) was undertaken during this season, consisting of 3,150 trap days over a ten-week period (Table 5). During this survey, all non-compliant rainwater tanks were positive (three of three) for *Ae. aegypti*, while three compliant tanks were negative (Table 5). Surveillance during the second suppression program (2015–2016), that included a rainwater tank survey for compliance and background *Ae. aegypti* population monitoring, revealed 46.6% of tanks non-compliant with regulatory standards [37]. Wild (unmarked) *Ae. aegypti* were collected in twelve of 26 premises (46%), while four of ten rainwater tanks sampled were positive (Fig 3). Control activities for the second suppression program (2015–2016) included treating 40 containers and one tank with S-methoprene pellets and briquettes, respectively, with four IRS treatments, 24 non-compliant tanks sealed (Fig 4), and 44 rubbish containers removed (Table 2).

Surveillance activities for 2016–2017 summer season are unavailable, however, this was the first season that BRC reported *Ae. aegypti* to be absent from Gin Gin. Likewise, a five-week BRC survey with five BGS traps in five premises during the 2017–2018 summer season

**Table 4. Pairwise FST estimates for eleven samples of *Aedes aegypti* from Queensland, Australia.** Bold indicates no significant differentiation (Table modified from Rašić *et al*. [35] to include data from Gin Gin).

|  | 1 | 2 | 3 | 4 | 5 | 6 | 7 | 8 | 9 | 10 | 11 |
|---|---|---|---|---|---|---|---|---|---|---|---|
| **1 Yorkeys Knob** | 0 | | | | | | | | | | |
| **2 Gordonvale** | **0.0126** | 0 | | | | | | | | | |
| **3 Ingham** | 0.0420 | 0.0507 | 0 | | | | | | | | |
| **4 Rockhampton** | 0.0458 | 0.0272 | 0.0831 | 0 | | | | | | | |
| **5 Mt Morgan** | 0.0786 | 0.0789 | 0.1191 | 0.0283 | 0 | | | | | | |
| **6 Duaringa** | 0.1117 | 0.1012 | 0.1589 | 0.0423 | 0.0499 | 0 | | | | | |
| **7 Bluff** | 0.0619 | 0.0759 | 0.0683 | 0.0871 | 0.1174 | 0.1541 | 0 | | | | |
| **8 Emerald** | 0.0548 | 0.0734 | 0.0972 | 0.0527 | 0.0616 | 0.1093 | 0.1333 | 0 | | | |
| **9 Capella** | 0.1131 | 0.1263 | 0.1172 | 0.0910 | 0.1028 | 0.0611 | 0.0970 | 0.1533 | 0 | | |
| **10 Longreach** | 0.0649 | 0.0708 | 0.1277 | 0.0567 | 0.0651 | 0.0573 | 0.1599 | 0.0735 | 0.1215 | 0 | |
| **11 Gin Gin** | 0.0779 | 0.1134 | 0.1785 | 0.1008 | 0.1455 | 0.1588 | 0.1695 | 0.1119 | 0.1691 | 0.1130 | 0 |

revealed *Ae. aegypti* to be absent (Fig 3). During the 2018–2019 season both the BRC and the Commonwealth Scientific and Industrial Research Organisation (CSIRO) undertook trapping surveys at ten premises over a ten-week period, representing 1,365 trap days with both BGS and ovitraps, with no *Ae. aegypti* detected (Table 5 and Fig 3).

Enhanced RSVP ovitrap surveillance (2019–2020) did not detect *Ae. aegypti* in any of the blocks where *Ae. aegypti* had been present previously (Fig 3). A total of 21 premises were surveyed across 13 different blocks for a total of 1,176 trap days across eight weeks (Table 5). A summary of blocks positive to *Ae. aegypti* previous to 2017 suggest the species distribution was primarily in central areas within the town (Fig 3), while surveillance effort of 3,591 trap days during the period from 2017 to 2020 suggests the species is no longer present in high-risk central blocks (Table 5 and Fig 3 2017–2020). A timeline of all control and surveillance activities can be found in Fig 5.

**Table 5. Summary results of *Aedes aegypti* surveillance activities in the town of Gin Gin, Australia.** Trap days indicate the number of traps placed in the town multiplied by the number of days each was sampled during a summer season (November-May).

| Summer Season | Survey Type | Premises Surveyed | Positive Premises (%) | Survey Period (days) | Trap Days | Positive Traps (%) |
|---|---|---|---|---|---|---|
| 2012–2013 | Larval | na | 14 (na) | 2 | na | na |
| 2013–2014 | Larval | 73 | 5 (7) | 2 | na | na |
| | GAT | 6 | 1 (17) | 13 | 78 | 1 (17) |
| 2014–2015 | Ovitrap | 9 | 4 (44) | 70 | 2,520 | 6 (3) |
| | GAT | 9 | 2 (22) | 70 | 630 | 5 (11) |
| | Rainwater Tank | 7 | 3 (43) | 70 | na | na |
| 2015–2016 | BG Trap | 26 | 12 (46) | 13 | 455 | 17 (49) |
| | GAT | 26 | 2 (8) | 13 | 910 | 10 (14) |
| | Rainwater Tank | 10 | 4 (40) | 1 | na | na |
| 2016–2017 | Data Unavailable | | | | | |
| 2017–2018 | BG Trap | 5 | 0 (0) | 35 | 175 | 0 (0) |
| 2018–2019 | BG Trap | 10 | 0 (0) | 105 | 560 | 0 (0) |
| | Ovitrap | 6 | 0 (0) | 70 | 1,680 | 0 (0) |
| 2019–2020 | Ovitrap (RSVP) | 21 | 0 (0) | 56 | 1,176 | 0 (0) |

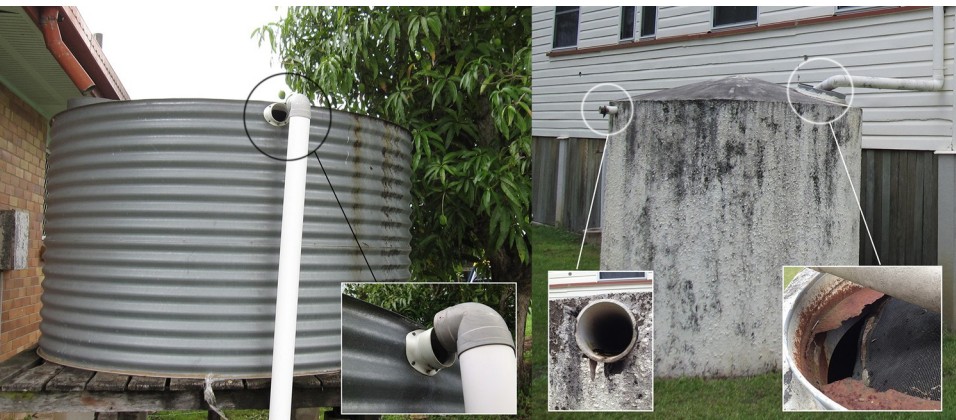

**Fig 4. Non-compliant rainwater tanks (with anti-mosquito regulations) identified in Gin Gin containing *Aedes aegypti*.** Inserts show detailed views of exposed overflows and rusted inflow sieves.

## Discussion

In Australia, the prevention of exotic vector-borne disease is a public health matter of national importance. A key component to understand disease transmission risk is access to data of the current distribution and abundance of vector species within different spatio-temporal scales, that range from local contact case addresses, larger environs of town or city and regional perspectives. Additionally, contemporaneous surveillance in regions that are vulnerable to stochastic invasion by urban vectors is required to enable the timely triggering of elimination campaigns as a strategy to avoid scenarios where cryptic outbreaks result from the belated recognition of covert incursions by vectors. However, the logistical challenges to obtain these data and perform elimination protocols is significant. In 2011, *Ae. aegypti* was re-detected in Gin Gin, a small regional town on a major highway into SEQ (a region which contains ~70% of

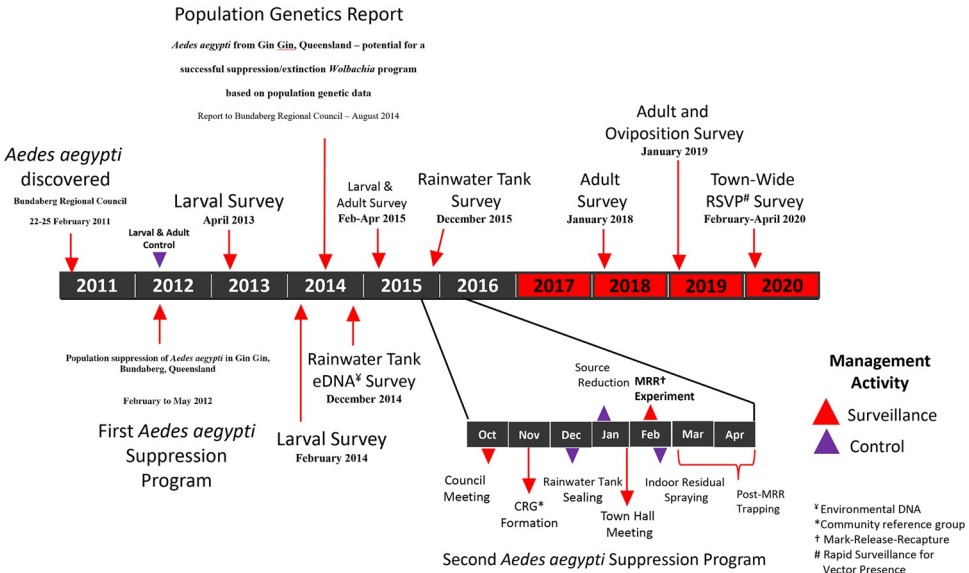

**Fig 5. Timeline of surveillance and control activities which led to the elimination of *Aedes aegypti* from Gin Gin, Australia.** Black and red bars indicate the period when *Ae. aegypti* was surveyed as present and not present, respectively.

Queensland's population). Infestations in towns on the margins of SEQ, particularly those that are also located on major transport pathways, increase the risk that *Ae. aegypti* could re-invade major population centres such as Brisbane. Gin Gin provides a case study of the sustained and concerted public health effort, involving both traditional mosquito control and innovative entomological surveillance 2012–2020, that is required to obtain confidence that an *Ae. aegypti* population was suppressed below the level of detection in a small town. This important public health outcome demonstrates that traditional mosquito control is effective at suppressing and potentially eliminating *Ae. aegypti* populations in small towns. Similar techniques that targeted larval habitat were first utilized to control *Ae. aegypti* during the construction of the Panama Canal [49] and later when the species was eliminated from 23 countries in the Americas [50,51]. However, lessons learnt since these historical campaigns suggest traditional methods may not be sustainable for larger, modern towns and cities, where introductions will vary both spatially (heterogeneous distributions resulting from low dispersal behaviours) and temporally (drought-resistant eggs) [52,53]. Unfortunately, the current invasions of *Ae. aegypti* through California and *Ae. albopictus* through Europe are examples where undetected introductions were able to escape modern control methods [54,55]. Elimination from cities in Australia and elsewhere will be particularly difficult without investment in national capacity and capability to perform large scale interventions. The logistical resources and costs to scale our model to large urban areas would be significant [56] and suggests strategic planning for incursions [20] should embed genetic analyses and additional innovative measures within routine entomologic surveillance and emergency non-insecticide-based responses, including *Wolbachia* or other emergent technologies, should be considered in public health policy.

The Gin Gin elimination documented here benefited from the local support of health authorities and scientists partnering to conduct regular entomological surveillance and control activities. During the second *Ae. aegypti* suppression program in 2015/2016, the informal technical advisory group had oversight of key mosquito control activities which integrated source reduction (sealed rainwater tanks) and treatment of larval habitat with juvenile hormones (S-methoprene) in the central business and residential areas of the town. This effectively suppressed the population to levels below the sensitive detection thresholds set by modern surveillance methods such as RSVP. By late summer 2019/2020 and after four years of surveillance activities, *Ae. aegypti* was not detected in Gin Gin. The 2015/2016 intervention was the culmination of sustained and concerted effort by public health authorities and the community to destabilize the *Ae. aegypti* population. This effort included several local initiatives:

1. Effective engagement with the local community, which were highly supportive of mosquito surveillance activities, and ensured ongoing compliance with health authorities;

2. Ongoing surveillance that identified key rainwater tanks acting as major urban mosquito population sources; and

3. Consistent pressure/focus from local government that identified rainwater tanks non-compliant with mosquito regulations were drained, sealed or removed which removed egg banks and ensured larval habitat was unavailable during periods of low rainfall.

*Aedes aegypti* was initially re-detected in a single block in northern Gin Gin during a routine larval survey. Interestingly, two similar surveys had not previously identified the species, suggesting the population may have persisted at very low levels or been recently re-introduced. Such uncertainty highlights the logistical challenges of traditional house-to-house, presence-absence surveillance for urban mosquitoes that can persist for extended periods as drought-resistant eggs. This species exhibits low movement over a lifetime (<200 m), however, *Ae. aegypti* and other anthropophilic species utilise human-mediated transportation thereby

facilitating long-distance dispersal [57,58]. This long-distance dispersal may be one factor that contributed to the first detection of the species within the northern area of the town. Several reasons could be hypothesised for the 2011 re-detection of the species. The positive residential block contains the local showground which hosts a constant flow of travellers who overnight in campervans. There is also a large commercial trucking stop at the northern end of Gin Gin, two blocks from the detection where large numbers of trucks stay overnight after travelling for extended periods from areas where *Ae. aegypti* is abundant. Adult mosquitoes may have entered a campervan or truck freight and 'hitchhiked' from northern or central Queensland to Gin Gin, a trait which is common in anthropophilic mosquitoes [59]. Alternatively, the detection is a remnant historical population last detected in routine house-to-house surveys in 1986 (Brian L. Montgomery, Queensland Health, pers. comm.). The determination of point-of-origin requires access to high resolution, genomic sequencing techniques and a separate analysis is currently being undertaken on *Ae. aegypti* populations in the region [45].

Genetic analysis indicated that the *Ae. aegypti* within Gin Gin formed an homogeneous cluster with a small effective population size. The resolution of the analysis did not establish whether this population was newly founded in 2011 or a relict population, but no first-generation immigrants were identified. The population was deemed vulnerable to elimination with low potential for reinvasion given the very low level of genetic admixture observed as well as the level of inbreeding and the degree of differentiation from the other samples from Queensland. A similar conclusion was reached for several semi-isolated populations in other parts of central Queensland [35]. A small effective population size in *Ae. aegypti* has been observed in other locations around the world and is a favourable attribute for population suppression or replacement of this species [60]. Genetic and genomic analysis of any *Ae. aegypti* incursion (e.g. Tennant Creek invasion of 2021) [13,14] is important to identify invasion pathways and determine the long-term effectiveness of suppression strategies [58,61]. Such analyses, based on reference populations throughout Queensland, would provide definitive answers to whether future detections in Gin Gin are from the original population or introduced from elsewhere. Characterization of *Ae. aegypti* genotypes from all Queensland population centres would inform a point-of-origin assessment within SEQ and would be useful for identifying or eliminating sources of other Australian incursions. Genomic databases of *Ae. aegypti* from southeast Asia and the Indo-Pacific region would also provide a reference point for potential international incursions [61].

Modern elimination campaigns that have used traditional forms of urban mosquito control, typically involving community education, source reduction, and residual insecticides, have been effective at eliminating *Ae. aegypti* populations from isolated towns in regional Australia. For example, the first elimination campaign in Australia removed *Ae. aegypti* from Brisbane and surrounding areas during the mid-twentieth century when the city was much smaller and artificial containers were rare [5]. It has been suggested the species was eliminated from Brisbane via effective anti-mosquito regulations which targeted larval habitat such as unsealed rainwater tanks [5,8]. More recently, the species has been eliminated from the small and isolated communities of Groote Eylandt [13] and Tennant Creek [14] in the Northern Territory. It is likely that Australia's low rainfall contributed to the long-term and permanent suppression of those populations [62]. Our findings suggest the removel of key larval habitat such as unsealed rainwater tanks, contributed to the dissapearance of *Ae. aegypti* from Gin Gin.

A novel method of population suppression that exploits low rainfall conditions is the application of *Wolbachia* to utilize the deleterious effects of certain strains [45,63]. This strategy utilizes the loss of desiccation resistance in *Aedes* eggs to eliminate a population over extended dry periods. This 'replace and suppress' strategy would not only prevent dengue transmission but is likely to be highly effective for suppressing populations in large urban settings that will

otherwise prove difficult logistically to inspect and treat with insecticides, exacerbated by non-treatment of cryptic larval habitat [64]. Utilizing the wet-dry seasonal dynamic of Australian landscapes will be important to future campaigns which seek to eliminate populations of container-inhabiting vectors like *Ae. aegypti* and similar techniques could be used in other dry areas where these species are established.

In Queensland, the prevention of dengue and control of vector species is the shared responsibility of both state and local government organisations. Queensland Health has the overall responsibility under the *Public Health Act 2005* [22] for the surveillance and control of communicable diseases in Queensland, including exotic mosquito-borne diseases such as dengue fever. Provisions within Chapter 2 of the *Public Health Act 2005* [22] provide local governments with the statutory support and powers to undertake mosquito surveillance and control activities (via insecticide treatment) and to prevent and control public health risks in relation to mosquitoes within residents' premises. This involves the Queensland Health chief executive sanctioning an authorised prevention and control program when an area is likely to contain an infestation of a disease vector such as *Ae. aegypti* or risk of an outbreak of vector-borne disease. For example, unmaintained rainwater tanks can be made to comply with the *Public Health Regulation 2018* [23] and *Public Health Act 2005* [22] by local authorities, so they no longer function as larval habitat for *Ae. aegypti* or other mosquito species. These essential powers were drawn upon during the period 2011–2020 to ensure residents in Gin Gin did not continue to store or removed containers where *Ae. aegypti* was present. Importantly however, the logistical challenge for health authorities to access and eliminate mosquitoes in all homes and businesses during control activities in very large towns and cities using chemical models of elimination is immense. Thus, a central element of urban mosquito control is to raise awareness about the community's role to adopt behaviours that eliminate mosquito presence at home and in the workplace. Community engagement targeting *Ae. aegypti* has been a foundational component for suppressing dengue from regions such as Vietnam [65], Brazil [66,67], Singapore [68] and Northern Australia [11]. In Australia, concurrent investment is required to provide baseline monitoring programs that are more representative of the spatio-temporal parameters of urban mosquitoes to establish entomological confidence in a negative result for invasive species. Use of surveillance programs that increase throughput and sensitivity by the use of molecular diagnostic platforms (RSVP) and that can be linked to citizen science platforms (Mozzie Monitors [69] and Zika Mozzie Seeker [70]) can provide opportunities to further increase sampling frequency and site number will inform detection thresholds.

Establishing and maintaining community support in Gin Gin was essential to the success of urban mosquito control and dengue prevention initiatives. An effective engagement strategy sets objectives and defines the underlying activities that will best meet these objectives and those of the project. Utilizing support from council and health officers from the local community promoted and encouraged local acceptance, ownership of the project's goals, and facilitated entomologic surveillance and control activities. Trust is an important element of community engagement which must be established and maintained to ensure community support throughout the life of the intervention. The formation of a community reference group during 2015–2016, was essential to building trust during the intervention [68] when rainwater tanks were sealed, and insecticides utilized for suppression. A community reference group can provide a social licence to operate and facilitates the transfer of information from scientists or health authorities to the community or opportunities for community concerns to be voiced. Efforts to ensure a comprehensive engagement strategy can foster increased community acceptance, provide local support for activities and even some level of ownership as it promotes both enthusiasm within the community and adherence to personal behaviours [67,68]. Effective acts that reduce urban mosquito breeding sites in residential premises and significantly

reduce the vulnerability of individuals and community to invasive urban species and associated diseases [65].

## Conclusion

The extensive and locally supported public health efforts documented here demonstrate that an integration of traditional mosquito control, a small genetically isolated mosquito population and public engagement can eliminate *Ae. aegypti* from a small regional town. Replicating this model at appropriate spatial-temporal scales for large towns and cities may prove extremely difficult to sustain without incorporating innovative solutions that provide an early warning capability and assist in monitoring the efficacy and longevity of suppression activities. Removing vectors from a region is a strategic solution to preventing disease transmission and further population spread. Re-emergence of dengue in Rockhampton after 60 years is a reminder that wild-type populations of *Ae. aegypti* in Queensland still represent a risk of disease transmission. While the SEQ region is currently considered vector-free, the redetection of *Ae. aegypti* in Gin Gin demonstrates that there is cause for caution. Detection thresholds are insensitive throughout much of the region and stochastic incursions risks remain via freight connections with large, established *Ae. aegypti* populations, and increased interceptions at international First Ports of Entry (airports and seaports) [61]. This regional risk is heightened by over 300,000 rainwater tanks installed throughout SEQ [19] which are generally unmonitored by authorities and are approaching the end of their warranty periods. Engaging communities to participate in surveillance (citizen science) may also encourage broader awareness and adoption of personal behaviours that reduce availability of residential and commercial sites to urban mosquito species that will also reduce regional vulnerability to invasive species and associated risk of exotic diseases, particularly in regions with a high number of viraemic travellers and/or proximal to First Ports of Entry. Application of the Gin Gin model to large towns and urban cities of SEQ and Australia will require significant investment in national capacity and capability. Robust and contemporaneous urban mosquito surveillance programs are required that are expansive and sustainable to provide the level of sensitivity required to provide regional confidence that towns and cities are absent of vectors and sensitive enough to detect incursions (that may be focal for many years) relatively early. This capability is enhanced by linking surveillance methods to molecular diagnostic methods to develop genetic reference libraries to define species identification, point-of-origin and insecticide resistance. In turn, this investment will build essential experience and baseline monitoring data that will inform elimination strategies if invasive vectors such as *Ae. aegypti* or *Ae. albopictus* are detected in major Australian cities.

## Supporting information

**S1 Text. Queensland Health Report: Population suppression of *Aedes aegypti* in Gin Gin, Bundaberg, Queensland (redistributed through a Creative Commons Attribution 3.0 Australia licence).** Fig A. Cycle of dengue transmission Fig D. Map of Gin Gin showing properties surveyed (yellow) and properties positive for *Ae. aegypti* (red). Table A. Dengue notifications by place of acquisition for Queensland (2001–2011) Table B. Dengue notifications by place of acquisition for Bundaberg LGA (2001–2011) Table C. Categories of Barker Hudson et al. (1988) including examples Table D. Mosquito species found in container habitats and positive premises. Table E. Prevalence of water bearing containers and number positive for *Ae. aegypti*. (PDF)

**S2 Text. Community Engagement Plan Table A: Community Engagement Timeline.**
(PDF)

**S3 Text. Risk Management Strategy.** Table A: Hazard Analysis and Risk Assessment.
(PDF)

**S4 Text. Mosquito Suppression Plan.** Table A: Timeline of suppression events including time budgeted for activity Table B: Staff availability and required for suppression program.
(PDF)

**S1 Table. Supplementary information for genetic analysis.** Table A. Samples of *Aedes aegypti* larvae for genetic analysis taken from Gin Gin, Queensland during summer 2012/13. Table B. Number of genetic clusters (K) assigned by STRUCTURE in a sample of *Aedes aegypti* from Gin Gin, Queensland, Australia. Figure A. Genetic clusters assigned by STRUCTURE for samples of *Aedes aegypti* from Queensland, Australia.
(PDF)

# Acknowledgments

We would like to thank all current and past staff at the Bundaberg and North Burnett Regional Councils who assisted with mosquito surveillance, and community engagement. We would like to thank Giselle Parsons and Sue Schuler from the BRC for undertaking IRS spraying. We would like to acknowledge the initial survey work undertaken in 2011 by Pipi Mottram and Jarod Butler at Queensland Health. Finally, we would like to acknowledge the community of Gin Gin for their enthusiasm and support during surveillance and control activities undertaken within the town.

# Author Contributions

**Conceptualization:** Brendan J. Trewin, Tim P. Hurst, Nancy M. Endersby-Harshman, Greg J. Crisp.

**Data curation:** Brendan J. Trewin, Tim P. Hurst, Nancy M. Endersby-Harshman.

**Formal analysis:** Brendan J. Trewin.

**Funding acquisition:** Brendan J. Trewin, Tim P. Hurst, Nancy M. Endersby-Harshman, Greg J. Crisp.

**Investigation:** Brendan J. Trewin, Tim P. Hurst, Jason S. Gilmore, Nancy M. Endersby-Harshman, Greg J. Crisp.

**Methodology:** Brendan J. Trewin, Brian L. Montgomery, Tim P. Hurst, Nancy M. Endersby-Harshman.

**Project administration:** Brendan J. Trewin, Tim P. Hurst, Greg J. Crisp.

**Resources:** Jason S. Gilmore, Greg J. Crisp.

**Validation:** Brendan J. Trewin, Nancy M. Endersby-Harshman.

**Visualization:** Brendan J. Trewin.

**Writing – original draft:** Brendan J. Trewin, Brian L. Montgomery, Tim P. Hurst, Nancy M. Endersby-Harshman.

**Writing – review & editing:** Brendan J. Trewin, Brian L. Montgomery, Tim P. Hurst, Jason S. Gilmore, Nancy M. Endersby-Harshman, Greg J. Crisp.

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
