## [Decision Letter · Decision Letter 0]

9 Aug 2021

Dear Dr Trewin,

Thank you very much for submitting your manuscript "Extensive public health initiatives drive the elimination of Aedes aegypti from a town in regional Queensland: a case study from Gin Gin, Australia." for consideration at PLOS Neglected Tropical Diseases. As with all papers reviewed by the journal, your manuscript was reviewed by members of the editorial board and by several independent reviewers. In light of the reviews (below this email), we would like to invite the resubmission of a significantly-revised version that takes into account the reviewers' comments. 

We cannot make any decision about publication until we have seen the revised manuscript and your response to the reviewers' comments. Your revised manuscript is also likely to be sent to reviewers for further evaluation.

Sincerely,

Claire Donald

Associate Editor

Benjamin Althouse

Deputy Editor

Reviewer's Responses to Questions

**Key Review Criteria Required for Acceptance?**

**Methods**

-Are the objectives of the study clearly articulated with a clear testable hypothesis stated?

-Is the study design appropriate to address the stated objectives?

-Is the population clearly described and appropriate for the hypothesis being tested?

-Is the sample size sufficient to ensure adequate power to address the hypothesis being tested?

-Were correct statistical analysis used to support conclusions?

-Are there concerns about ethical or regulatory requirements being met?

Reviewer #1: The article is not a traditional research study, but instead documents an eradication campaign of Ae. aegypti from an area in Australia. The objectives could be made more clear and the methods expanded on. The author references methods described in other papers, but does not include brief abbreviated methods to aid the reader in understanding how the study was done. In my opinion, an improvement of the methods is needed to improve the paper. 

Reference to a previously published MRR study seems misplaced in the methods or its relevance was not explained well enough. It's relevance to the presented study needs to be made more clear. 

Methods on how larval surveillance was conducted/ tools that were utilized should be expanded on.

Reviewer #2: Needs summary of control activities

**Results**

-Does the analysis presented match the analysis plan?

-Are the results clearly and completely presented?

-Are the figures (Tables, Images) of sufficient quality for clarity?

Reviewer #1: Both the results and discussion would benefit from an additional figure (that could be combined and used for the two separate sections) that has a visual representation of the surveillance activities and results. Because this study was conducted over several years and the surveillance activities changed year to year, it is hard to follow without a visual representation. The same goes for the results. A timeline indicating what activities were conducted when AND when Ae. aegypti were detected and when they weren't would be very useful. 

Some of the figures in the appendix would be better utilized in the main manuscript to aid in telling the story. For example, the figures in S6, if they could be combined into a figure show a FANTASTIC visual representation of the progression of surveillance and the author's findings. It is wasted as an appendix and would greatly aid a reader in interpreting the written findings of the authors.

Reviewer #2: summarize control activities

**Conclusions**

-Are the conclusions supported by the data presented?

-Are the limitations of analysis clearly described?

-Do the authors discuss how these data can be helpful to advance our understanding of the topic under study?

-Is public health relevance addressed?

Reviewer #1: The conclusions are supported by the presented data. However, in the discussion section, the authors have a great deal of work to do relating their study to the research of others. The discussion is very weakly cited which I see as the greatest weakness of this manuscript. When the authors do reference other studies, it is almost exclusively for other studies conducted in Australia. There is a great deal of literature on eradication campaigns, community engagement studies for mosquito control, etc. This literature was not included and related to the current study. Therefore, the presented study cannot be considered in the context of previously conducted work. The discussion needs substantial improvement in this way.

Reviewer #2: see text below

**Editorial and Data Presentation Modifications?**

Reviewer #1: Major Revision - expansion of methods; improve clarity of results; improve discussion through expansion/ relation to previously published works; create timeline to aid in understanding methods and results; move figures from S6 to main body of the manuscript to improve readability.

Reviewer #2: (No Response)

**Summary and General Comments**

Reviewer #1: Overall, it is clear that the authors invested a great deal of effort into this eradication campaign and documenting it's progression. It clearly worked and the area is free of Ae. aegypti as a result. Because this was a SUCCESSFUL eradication campaign, communicating the process and findings effectively is all the more important so the methods can be utilized by others wishing to do the same. Additionally, relating this research to others may also help readers understand what successful eradication campaigns have in common and why others failed. With these changes, I believe this will be a very valuable piece of research.

Reviewer #2: This ms describes vector control and surveillance activities in a small regional town near Brisbane Queensland Australia,. In recent years there has been increased concern regarding reinvasion of the dengue vector Aedes aegypti into Brisbane, where this mosquito once caused large epidemics of the virus. You could argue that the ms is rather trivial, the simple elimination of Ae. aegypti from a country Australian town. Afterall, it has been done before, most recently published accounts by Whelan et al. for incursions into Tennant Creek in the NT. That said, the ms describes operational use of some novel surveillance tools and genetic analysis that will be of interest to readers. 

‘A few suggestions to improve the ms. 

• The ms could be shortened. The conclusions are just a repeat if what was said in the discussion. If you want to use it, use dot point of key messages. 

• The ms really needs summary table of control activities by year: no. premises surveyed, no. +ve, no containers treated or removed;

• The 2011 survey: was that organised by Pipi Mottram of QH? If so, she should be included as an author.

• Please cite relevant key used to ID captured mosquitoes, for both opvitraps and BG surveys.

• Line 167: I’d move ovitrap description to where they were first used (line 119).

• Ln 181: Please provide RWT mesh size, according to statute.

• Ln 232: “Mosquitoes were present…” all species? or just aegypti?

• Ln 361: your genetic analysis should tell you if population is relict or recent introduction.

Other minor comments on the attached pdf.

PLOS authors have the option to publish the peer review history of their article (what does this mean?). If published, this will include your full peer review and any attached files.

Reviewer #1: No

Reviewer #2: No
---

## [Decision Letter · Decision Letter 1]

12 Nov 2021

Dear Dr Trewin,

Thank you very much for submitting your manuscript "Extensive public health initiatives drive the elimination of Aedes aegypti (Diptera, Culicidae) from a town in regional Queensland: a case study from Gin Gin, Australia." for consideration at PLOS Neglected Tropical Diseases. As with all papers reviewed by the journal, your manuscript was reviewed by members of the editorial board and by several independent reviewers. The reviewers appreciated the attention to an important topic. Based on the reviews, we are likely to accept this manuscript for publication, providing that you modify the manuscript according to the review recommendations. 

Sincerely,

Benjamin Althouse

Deputy Editor

Reviewer's Responses to Questions

**Key Review Criteria Required for Acceptance?**

**Methods**

-Are the objectives of the study clearly articulated with a clear testable hypothesis stated?

-Is the study design appropriate to address the stated objectives?

-Is the population clearly described and appropriate for the hypothesis being tested?

-Is the sample size sufficient to ensure adequate power to address the hypothesis being tested?

-Were correct statistical analysis used to support conclusions?

-Are there concerns about ethical or regulatory requirements being met?

Reviewer #1: Revision is satisfactory.

Reviewer #2: needs a cost summary

**Results**

-Does the analysis presented match the analysis plan?

-Are the results clearly and completely presented?

-Are the figures (Tables, Images) of sufficient quality for clarity?

Reviewer #1: Revision is satisfactory.

Reviewer #2: See below

**Conclusions**

-Are the conclusions supported by the data presented?

-Are the limitations of analysis clearly described?

-Do the authors discuss how these data can be helpful to advance our understanding of the topic under study?

-Is public health relevance addressed?

Reviewer #1: Revision is satisfactory.

Reviewer #2: Sorry I missed this point on 1st review, but cost is critical. 

In the Author summary you state: “Here we document the successful elimination of a population of Ae. aegypti from Gin 35 Gin, Australia, through a low-cost but extensive public health initiative”

 Then in the discussion line 388 you state “The logistical resources and costs to scale our model to large urban areas would be significant (56)…”

But no costings of the initiative are given. So, can you provide a brief summary of the cost of the intervention, in labor and materials. 

And how they compare to the NT (Tennant Creek, Groyte Is). elimination.

**Editorial and Data Presentation Modifications?**

Reviewer #1: Revision is satisfactory.

Reviewer #2: no

**Summary and General Comments**

Reviewer #1: The manuscript in its current version has satisfied my comments from the original submission. The modifications that the authors have incorporated does better justice to the extensive efforts they have invested in their eradication campaign over the years. This amount of work is impressive and I believe other readers/ researchers will find a great deal of value in this manuscript.

Reviewer #2: (No Response)

PLOS authors have the option to publish the peer review history of their article (what does this mean?). If published, this will include your full peer review and any attached files.

Reviewer #1: No

Reviewer #2: No

Figure Files:

Data Requirements:

Reproducibility:

References

---

## [Editor Report · Decision Letter 2]

9 Feb 2022

Dear Dr Trewin,

We are pleased to inform you that your manuscript 'Extensive public health initiatives drive the elimination of Aedes aegypti (Diptera, Culicidae) from a town in regional Queensland: a case study from Gin Gin, Australia.' has been provisionally accepted for publication in PLOS Neglected Tropical Diseases.

Best regards,

Benjamin Althouse

Deputy Editor

Benjamin Althouse

Deputy Editor

---

## [Editor Report · Acceptance letter]

4 Apr 2022

Dear Dr Trewin,

We are delighted to inform you that your manuscript, "Extensive public health initiatives drive the elimination of Aedes aegypti (Diptera, Culicidae) from a town in regional Queensland: a case study from Gin Gin, Australia," has been formally accepted for publication in PLOS Neglected Tropical Diseases.

Best regards,

Shaden Kamhawi

co-Editor-in-Chief

Paul Brindley

co-Editor-in-Chief
